# Molecular mechanisms of Holliday junction branch migration catalyzed by an asymmetric RuvB hexamer

Anthony D. Rish [1,2,3,6], Zhangfei Shen [2,3,6], Zhenhang Chen [2,3,5], Nan Zhang[3,4], Qingfei Zheng [1,2,3,4] & Tian-Min Fu [1,2,3] ✉

The Holliday junction (HJ) is a DNA intermediate of homologous recombination, involved in many fundamental physiological processes. RuvB, an ATPase motor protein, drives branch migration of the Holliday junction with a mechanism that had yet to be elucidated. Here we report two cryo-EM structures of RuvB, providing a comprehensive understanding of HJ branch migration. RuvB assembles into a spiral staircase, ring-like hexamer, encircling dsDNA. Four protomers of RuvB contact the DNA backbone with a translocation step size of 2 nucleotides. The variation of nucleotide-binding states in RuvB supports a sequential model for ATP hydrolysis and nucleotide recycling, which occur at separate, singular positions. RuvB's asymmetric assembly also explains the 6:4 stoichiometry between the RuvB/RuvA complex, which coordinates HJ migration in bacteria. Taken together, we provide a mechanistic understanding of HJ branch migration facilitated by RuvB, which may be universally shared by prokaryotic and eukaryotic organisms.

Homologous recombination is an omnipresent cellular process that is critical not just for the maintenance of genomic stability after DNA damage but also for generating genetic diversity and has been associated with a variety of human diseases[1]. After DNA damage is detected, a series of steps, including strand invasion and D-loop formation, occurs before a key intermediate structure of homologous recombination, the Holliday junction (HJ), is formed[2]. The HJ consists of two double-stranded DNA (dsDNA) helices that are separated into four hetero-duplex strands through a crossover point. In bacteria, RuvA, RuvB, and RuvC are involved in processing the HJ. RuvA was shown to assemble into a tetramer for recognizing the HJ core and recruits RuvB. As a motor protein, RuvB can bind the dsDNA part of the HJ and promote the movement of HJ, known as branch migration. Though earlier studies showed that the RuvA-RuvB complex, but not RuvB alone, can unwind dsDNA in the presence of ATP in vitro[3], later studies revealed that RuvB drives branch migration by translocating along duplex DNA without unwinding dsDNA[4]. Eventually, the HJ is resolved back into two complete DNA strands by RuvC, an endonuclease. In eukaryotes, there are homologous proteins of the RuvAB complex, such as Rad54 in humans, to conduct HJ branch migration[5–8]. Therefore, fundamental mechanisms of HJ migration in bacteria may be universally shared by all organisms.

Earlier structural studies have provided rich information regarding interactions of RuvA with HJ DNA and RuvB. RuvA has been shown to consist of three distinct domains (Domain I, II, and III). A crystal structure of RuvA in complex with HJ has revealed that RuvA forms a unique tetrameric architecture, in which domains I and II are responsible for Holliday junction binding[9]. Domain III of RuvA was shown to directly interact with RuvB for the loading of RuvB onto HJ[10]. Crystal structures of RuvB from different bacteria showed that RuvB shares a conserved architecture that is composed of three domains: N-terminal domain (NTD), Middle domain (MD), and C-terminal domain (CTD)[11–13].

[1]The Ohio State Biochemistry Program, The Ohio State University, Columbus, OH 43210, USA. [2]Department of Biological Chemistry and Pharmacology, The Ohio State University, Columbus, OH 43210, USA. [3]Center for Cancer Metabolism, The Ohio State University Comprehensive Cancer Center, Columbus, OH 43210, USA. [4]Department of Radiation Oncology, College of Medicine, The Ohio State University, Columbus, OH 43210, USA. [5]Present address: Department of Biochemistry, Emory University School of Medicine, Atlanta, GA 30322, USA. [6]These authors contributed equally: Anthony D. Rish, Zhangfei Shen. ✉e-mail: fu.978@osu.edu

A feature β-hairpin in RuvB NTD was shown to interact with domain III of RuvA through hydrophobic interactions[10].

RuvB belongs to the AAA+ (ATPase associated with various cellular activities) family, which includes many members involved in numerous physiological processes[14]. All AAA+ proteins share a core domain with an αβα sandwich-fold, which contains two key motifs, the Walker A motif and the Walker B motif[15]. Walker A motif is composed of GXXXGK(T/S), where X stands for any amino acid, and the Walker B motif consists of hhhhD(D/E), where h represents any hydrophobic amino acids. The Walker A motif is responsible for binding nucleotides, and the Walker B motif coordinates $Mg^{2+}$ ions and water molecules to catalyze the hydrolysis of nucleotides. Structural studies on classical members of AAA+ ATPase, including Cdc48, p97, ClpX, N-ethylmaleimide-sensitive-factor (NSF), and FtsK, showed that AAA+ proteins tend to form ring-like structures for function[16–24]. Similarly, earlier low-resolution electron microscopy showed that RuvB assembles into a hexamer on HJ DNA, forming a tripartite complex by flanking a RuvA octamer on both ends[25,26]. However, the low-resolution structure of RuvA-RuvB-HJ limits a mechanistic understanding of Holliday junction branch migration.

Here, we present cryo-EM structures of RuvB in complex with dsDNA with resolutions up to 2.9 Å, showing that RuvB forms a spiral staircase to facilitate the HJ migration. During our manuscript preparation, similar structures of *Streptococcus thermophilus* RuvB in complex with RuvA and DNA were published in *Nature*[13]. The asymmetric hexameric assembly of RuvB, around dsDNA, provides a mechanistic understanding of Holliday junction migration, in which ATP hydrolysis drives conformational changes of RuvB, leading to the pulling and revolving of DNA inside the central pore of RuvB by two nucleotides per step. Moreover, our structural analysis also revealed that ATP hydrolysis occurs at the top position of the spiral staircase most probably in a sequential manner. Additionally, the asymmetric assembly of RuvB also indicates a potential asymmetric engagement with RuvA and an association-and-detachment mechanism for the interactions between RuvA and RuvB during HJ branch migration. Together, we propose that HJ migration includes a series of highly coordinated events, including nucleotide cycling, RuvB conformational changes that trigger DNA pulling and revolving, as well as the association and detachment of RuvA to RuvB.

## Results
### Overall structures of the RuvB-DNA complex
To gain a mechanistic understanding of Holliday junction branch migration, we expressed and purified *Thermus thermophilus* RuvA and RuvB from *Escherichia coli*. Then, we incubated RuvA and RuvB with DNA substrates in the presence of 2 mM ATPγS to assemble the HJ complex. We also incubated RuvB with DNA substrates and 2 mM ATPγS to assemble the RuvB-DNA complex, which was further purified by gel filtration (Supplementary Fig. 1a, b). Gel filtration showed that RuvB alone may form a monomer while the RuvB and DNA assemble into a large complex (Supplementary Fig. 1a, b). The purified samples were applied to cryo-EM grids for screening and data collection using an FEI Titan Krios equipped with a K3 detector (Supplementary Fig. 1c). As we failed to generate useful datasets for our HJ complex containing RuvA/RuvB/DNA, we focused on elucidating the assembly of the RuvB-DNA complex. Data processing in cryoSPARC resulted in two maps with overall resolutions of 2.97 Å and 3.20 Å (Fig. 1a, b, Supplementary Figs. 2, 3a–d, Supplementary Table 1). These high-resolution maps allowed us to build atomic models of the RuvB-DNA complexes, denoted as RuvB hexamer and RuvB dodecamer models (Fig. 1c, d, Supplementary Fig. 4 and Supplementary Table 1), respectively. The RuvB dodecamer is formed by two RuvB hexamers in a head-to-head manner with approximate dimensions of 158 Å × 127 Å × 127 Å (Fig. 1a, c). The RuvB hexamer has approximate dimensions of 80 Å × 127 Å × 127 Å (Fig. 1b, d). RuvB assembles into a ring-like hexamer with dsDNA in the central pore that

directly contacts four subunits (Fig. 1c, d). The RuvB hexamer model resembles RuvB hexamers from the dodecamer with minor differences. Notably, a feature β-hairpin in the NTD of RuvB is invisible in most protomers of RuvB hexamer but is well-defined in all the protomers of RuvB dodecamer (Fig. 1e). Similar differences were also observed between our RuvB hexamer and *S. thermophilus* RuvB hexamer (Fig. 1f). In the RuvB dodecamer, we observed interactions mediated by the feature β-hairpins of RuvB (Fig. 1c), whereas this feature β-hairpin interacts with the C-terminal domain of RuvA in *S. thermophilus* RuvB. These interactions, which are absent in our *T. thermophilus* RuvB hexamer, may stabilize the conformations of the feature β-hairpin in our RuvB dodecamer and in *S. thermophilus* RuvB. Moreover, the overlaid structures of our RuvB dodecamer and *S. thermophilus* RuvB showed obvious conformational changes with respect to the feature β-hairpins of equivalent protomers of RuvB (Fig. 1g). Together, these structural comparisons suggest that the feature β-hairpin of RuvB is conformationally flexible, which may be critical for its unique interaction mode with RuvA in the Holliday junction complex (See section on asymmetric interactions with RuvA).

Under physiological conditions, RuvB hexamer flanks two ends of RuvA-HJ DNA to catalyze the migration of HJ. As such, the RuvB dodecamer we obtained is not biologically relevant. Moreover, the RuvB hexamer, in complex with DNA, is sufficient to reveal the mechanism of HJ branch migration. Therefore, we will focus all further discussions on the RuvB hexamer model in our manuscript.

### Assembly of the RuvB hexamer
Six RuvB protomers assemble into an asymmetric ring-like hexamer with a 3.0-3.4 nm-wide central pore that accommodates duplex DNA (Figs. 1d, 2a). As reported before, each protomer of RuvB is composed of an NTD, an MD, and a CTD (Fig. 2b, Supplementary Fig. 5). Strikingly, each protomer (A to F) in the RuvB hexamer adopted a unique conformation (Fig. 2c). We aligned the RuvB NTDs of B to F with that of the target protomer A. The relative orientations of RuvB CTDs display clear deviations between protomers with rotational angles ranging from 5° to 23° relative to the Z axis (Fig. 2c). In contrast, the separated individual domains of RuvB from each protomer of RuvB hexamer aligned well (Supplementary Fig. 6a–c). Together, these data suggest that rigid body motions occur among the three domains of each individual RuvB protomer during the assembly of the RuvB-DNA complex.

Compared to the previously determined crystal structure of RuvB (PDB code: 1HQC), protomers A, B, C, and D in the RuvB hexamer display large conformational changes, in which the CTDs rotate away from the NTDs (Fig. 2d, Supplementary Fig. 6d). In contrast, protomer E is almost identical to the crystal structure of RuvB while protomer F displays small conformational variations (Fig. 2d and Supplementary Fig. 6d). Interestingly, protomers A, B, C, and D are involved in DNA binding while E and F are DNA-disengaged protomers, suggesting that DNA engagement may trigger conformation changes of RuvB protomers from a closed to a more open conformation (Fig. 2e). Thus, we propose that the assembly of RuvB-DNA complex may be a hierarchical process involving multiple sequential events, in which one protomer of RuvB binds to DNA followed by conformational changes of RuvB and the cooperative assembly of additional RuvB-RuvB interactions (Fig. 2e).

The conformational diversity between RuvB protomers not only generates the asymmetric RuvB ring but also results in large variations among protomer interfaces (Fig. 3a and Supplementary Fig. 7a–f). The interface between protomers B and C and the interface between protomers B and A are most extensive with buried areas of 3,860 Å² and 3,721 Å², respectively; the interface between protomers D and E, and protomers D and C are less extensive with buried areas of 2,508 Å² and 2,256 Å², respectively; the interface between protomers E and F and the interface between protomers A and F are least extensive with buried areas of 1,562 Å² and 1,537 Å², respectively (Fig. 3a). Detailed examination of interfaces between neighboring RuvB protomers

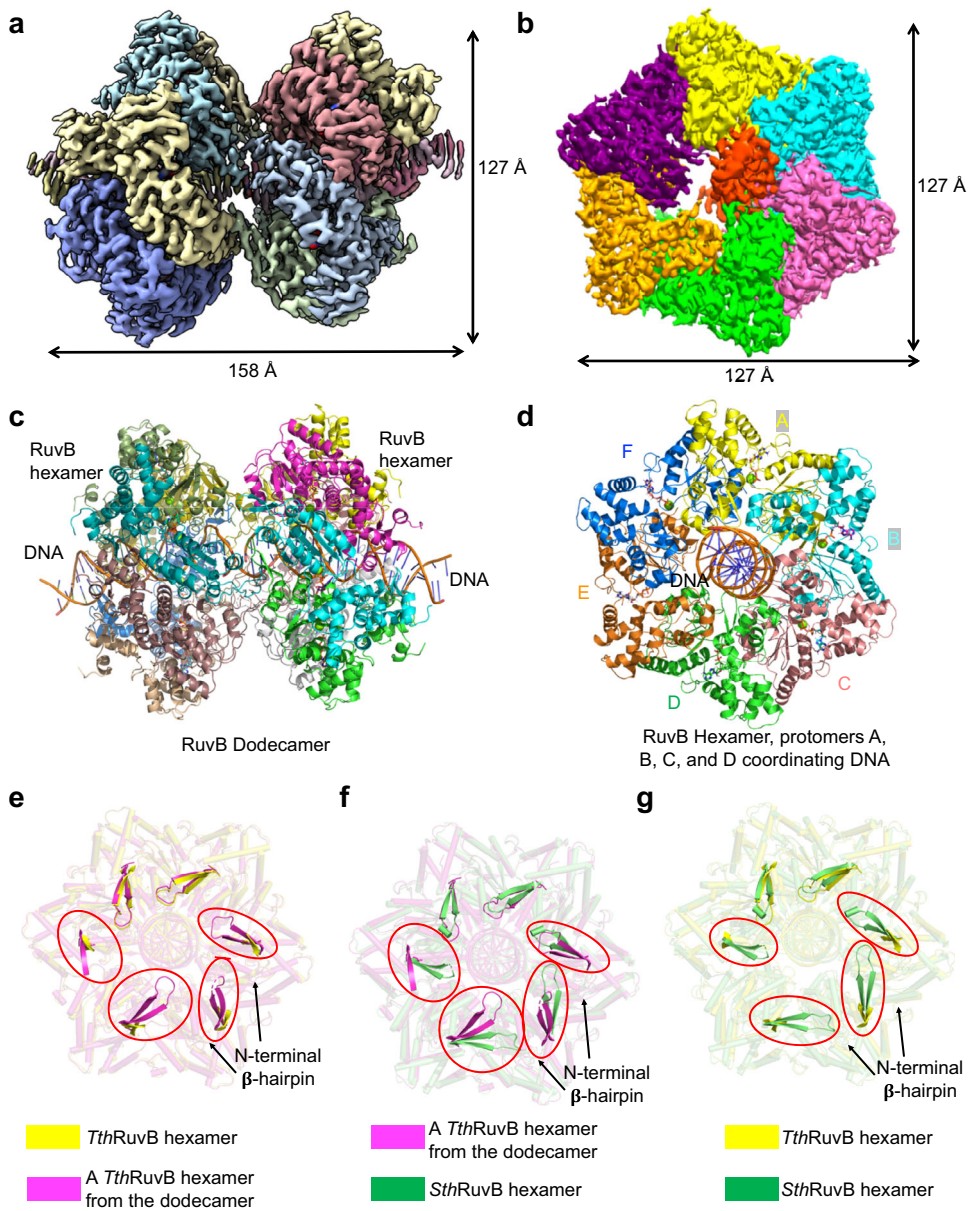

**Fig. 1 | Overall structures of the RuvB-DNA complex. a** The cryo-EM density map of RuvB dodecamer in complex with dsDNA with each subunit colored individually. The dodecamer has estimated dimensions of 158 Å by 127 Å by 127 Å. **b** The cryo-EM density map of RuvB hexamer in complex with dsDNA with each subunit colored individually. The hexamer has estimated dimensions of 80 Å by 127 Å by 127 Å. **c** Ribbon diagrams of RuvB dodecamer with dsDNA in the central pore with each subunit colored as in **a**. **d** Ribbon diagrams of RuvB hexamer with dsDNA in the central pore with protomers A (yellow), B (cyan), C (salmon), D (green), E (orange), F (blue) colored differently. Nucleotides are shown as sticks. **e–g** Structural comparisons of *T. thermophilus* RuvB hexamer (yellow), *S. thermophilus* hexamer (green, PDB 7PBS), and a *T. thermophilus* hexamer from the dodecamer (magenta), revealing the conformational dynamics of a feature β-hairpin in the N-terminal domain of RuvB.

showed that all three domains of one protomer pack tightly with domains of its partner in the AB dimer and BC dimer, whereas the interactions mediated by domain N are either reduced or abolished in the CD dimer, DE dimer, EF dimer, and AF dimer (Fig. 3b–d and Supplementary Fig. 7a–f). Despite these differences, all the interfaces between RuvB protomers are dominated by charge-charge interactions (Fig. 3b–d). For example, in the BC dimer, D277 (C) and R147 (B), D216 (C) and R34 (B), R215 (C) and E39 (B), R212 (C) and E39 (B), R205 (C) and E115 (B), and R7 (C) and D116 (B) all form salt bridges to glue the protomers B and C together (Fig. 3e).

## DNA binding by RuvB
Duplex DNA density is clearly defined for more than 20 bp, which traverses the central pore of RuvB hexamer from one end to the other

(Fig. 4a, b). Among the six protomers of the RuvB hexamer, four protomers (A, B, C, and D) assemble into a 'spiral staircase' for coordinating dsDNA while the other two protomers (E and F) close the spiral staircase without interacting with DNA substrate (Fig. 4a, b). Notably, different from the Ftsk hexamer that only interacts with one strand of dsDNA[23], both strands of the DNA substrate interact with RuvB protomers A, B, C, and D (Fig. 4c & Supplementary Fig. 8a). Specifically, three conserved arginine residues R297, R300, and R302 in the RuvB CTD form a motif known as an arginine finger to interact with the negatively charged DNA backbone (Fig. 4c, Supplementary Fig. 8a and 5). Four arginine fingers from the DNA-interacting protomers were nicely positioned in a helical path to follow the DNA's helix precisely and established strong interactions with both strands of the duplex DNA substrate. Moreover, we observed a second

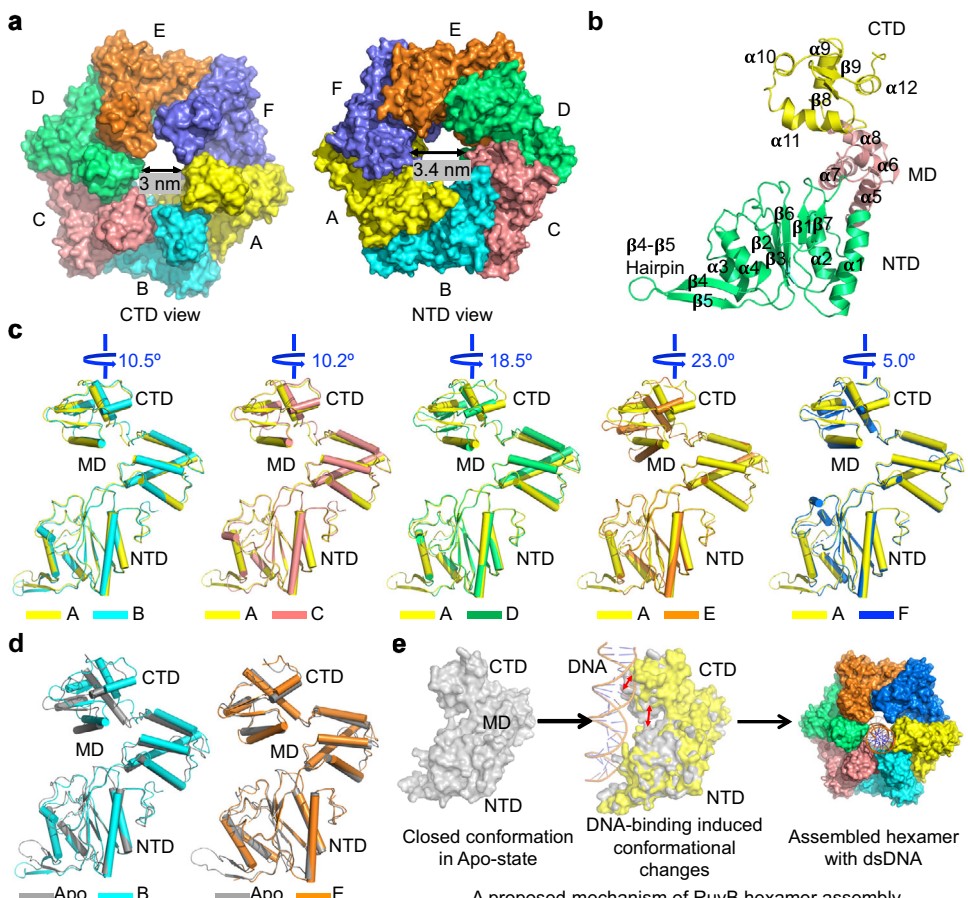

**Fig. 2 | The assembly of RuvB hexamer. a** Surface representation of RuvB hexamer with each protomer colored individually, A (yellow), B (cyan), C (salmon), D (green), E (orange), F (purple), showing the diameters of its central pore. **b** Ribbon diagram of a RuvB protomer with secondary structural elements labeled, CTD (yellow), MD (salmon), NTD (green). **c** Overlaid structures of RuvB protomers with A in yellow, B in cyan, C in salmon, D in green, E in orange, and F in blue, revealing the conformational differences of the six protomers in RuvB hexamer. **d** Structural comparisons of RuvB protomers B (yellow), E (orange), and the crystal structure of RuvB (gray). **e** Diagrams illustrating a potential assembly mechanism of RuvB hexamer in complex with dsDNA.

DNA-engagement site in our structure, which was not seen in the *S. thermophilus* structure, perhaps due to poor DNA density at the corresponding site. R101 next to the Walker B motif and R104 in the β6-β7 loop, two highly conserved residues across species, engaged the backbone of one strand of the dsDNA (Fig. 4d, Supplementary Figs. 8b and 5). As all the DNA-engaging arginine residues interact with the phosphate backbone, RuvB can bind to any DNA substrates without sequence preference. Notably, we also noticed that in both DNA-engagement sites, the DNA-interacting arginine residues from protomers B and C are closer to the phosphate backbone than those from protomers A and D, suggesting that protomers B and C engage with DNA more tightly than protomers A and D (Supplementary Fig. 8a, b). In addition, the DNA-engaged arginine residues form a repeated binding pattern along one strand of dsDNA separated by the distance of two nucleotides (about 7 Å) (Fig. 4c–e), suggesting that the DNA substrate migrates about 7 Å at each step catalyzed by RuvB.

## Nucleotide cycling

Adenine nucleotides were observed to be located at the interface of two adjacent subunits and are exclusively nested in a cleft formed by domains RuvB NTD and RuvB MD of one RuvB protomer in *cis* (Fig. 5a, b). Consistent with a previous observation in the crystal structure of RuvB, the adenine base lies in a hydrophobic cleft, which is composed of Y14, I15, Y168, and the side chain of R179 (Fig. 5a, b)[11]. The phosphate groups are coordinated by R7, R205, and the conserved Walker A motif, specifically K51 (Fig. 5a and

Supplementary Fig. 5). R158 from the adjacent subunit in *trans* also interacts with the nucleotide phosphate groups (Fig. 5a). A magnesium ion was observed to be in proximity to the γ-phosphate and was coordinated by the conserved Walker B motif residues D97 and E98. (Fig. 5a and Supplementary Fig. 5). To our surprise, the six protomers of RuvB hexamer bind different adenine nucleotides (Fig. 5c). Protomers B, C, and D bind ATP molecules while protomers A, F, and E coordinate ADP molecules (Fig. 5c). Among the four DNA-engaging protomers, both protomers B and C contain ATP molecules based on the nicely fitted nucleotide density (Supplementary Fig. 9a, b). Notably, protomer A, located at the top of the staircase, might contain either ADP or ATP with higher occupancy for ADP since ADP can be nicely fitted into the density with a sigma value of 1.8, while ATP can also be nicely placed into the density when we raised the sigma value to 2.5 (Fig. 5d). This observation suggested that ATP hydrolysis occurs at the top of the staircase between protomers A and F. After ATP hydrolysis, the ADP molecules remain associated with RuvB at positions F and E (Supplementary Fig. 9c). To rule out whether the ADP molecules in protomers E and F are endogenous or generated by ATPγS hydrolysis, we did native mass spectrometry analysis of our purified RuvB and found that the majority of our purified RuvB binds endogenous ADP molecules (Supplementary Fig. 9d). Thus, the ADP molecules in protomers E and F were not exchanged into ATPγS during our assembly of the RuvB-DNA complex, suggesting that RuvB protomers at positions E and F prefer to bind ADP over ATPγS. The ADP

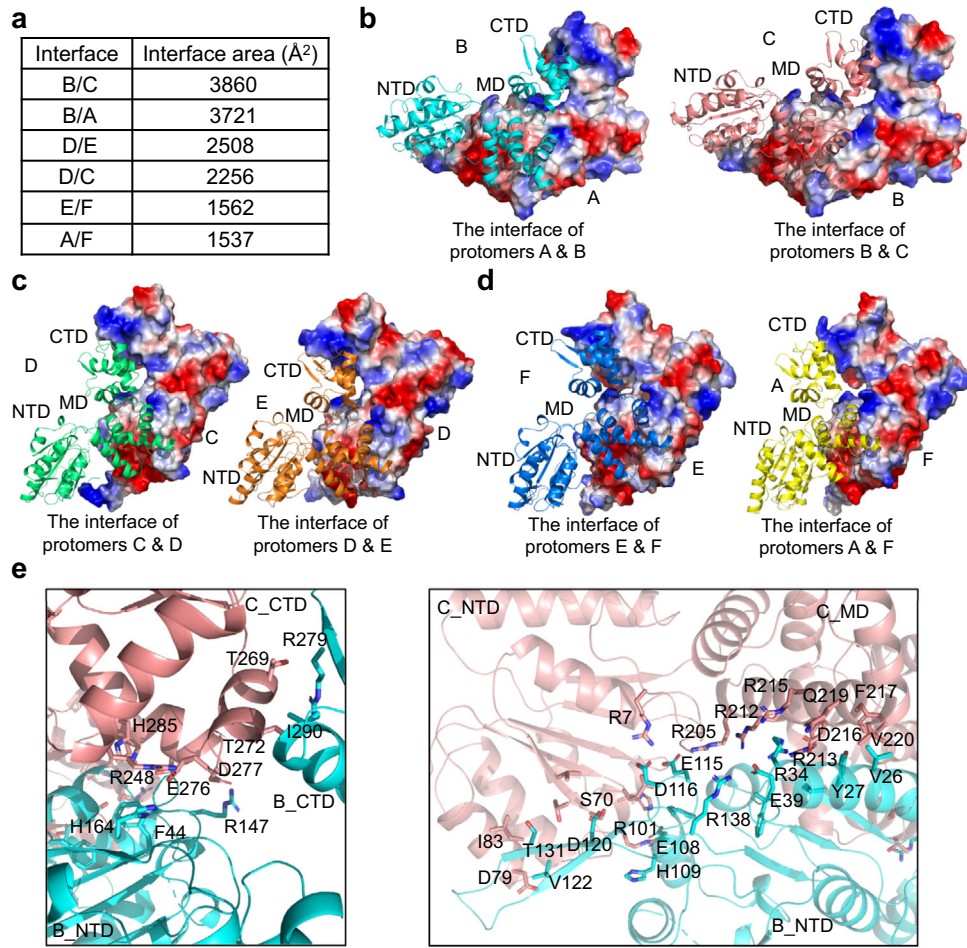

**Fig. 3 | Interfaces of RuvB dimers. a** Buried interface areas of the six dimers in RuvB hexamer. **b** The interfaces of the AB dimer and the BC dimer with the electrostatic surface representation of protomers A and B and ribbon diagrams of protomers B (cyan) and C (salmon), respectively. **c** The interfaces of the CD dimer and the DE dimer with electrostatic surface representation of protomers C and D and ribbon diagrams of protomers D (green) and E (orange), respectively. **d** The interfaces of the EF dimer and the AF dimer with the electrostatic surface representation of protomers E and F and ribbon diagrams of protomers F (blue) and A (yellow), respectively. **e** Detailed interaction between protomers C (salmon) and B (cyan) with interface residues highlighted as sticks. Electrostatic surface representations are colored by charge, with positively charged regions in blue and negatively charged regions in red.

to ATP exchange may occur at the position of protomer D as the nucleotide density in protomer D is very poor, likely due to the low occupancy of nucleotides at this position (Fig. 5e). Thus, ATP hydrolysis and exchange are spatiotemporally separated in the asymmetric hexamer of RuvB. Furthermore, this observation also indicates that nucleotide exchange and ATP hydrolysis are two discrete steps coordinated by the interfaces of protomers ED and AF, respectively.

To further understand why ATP hydrolysis occurs in protomer A but not others, we analyzed the interfaces of RuvB neighboring protomers. To our surprise, residues R158, E115, and D116 in a *trans* (adjacent) protomer, which are critical for ATP hydrolysis, are not positioned well for catalysis in the AF dimer but primed for catalysis in the CB dimer and the BA dimer (Fig. 5f–h and Supplementary Fig. 9e, f). Specifically, R158, which is involved in stabilizing the conformation of the **γ**-phosphate of ATP, is far away from γ-phosphate in the AF dimer; E115 was predicted to but did not form a salt bridge with R205 in the AF dimer; D116 also pointed away from the catalytic center in the AF dimer (Fig. 5f, g). These results suggested that the conformation of the AF dimer we captured may represent a post-hydrolysis state. In contrast, the conformations of the BA and the BC dimers represent a primed-catalysis state. This raised the question as to why ATP hydrolysis cannot occur at positions B and C. We speculate that the extensive interaction interfaces in the BA dimer and the BC dimer may limit

conformational changes within the N domain, which is critical for ATP hydrolysis. Additionally, the tight interfaces may also limit the free access of water molecules to the catalytic centers of the BA and BC dimers. Consistent with this speculation is that we observed unmodelled densities in the catalytic center of the AF dimer (Fig. 5i), which may be attributed to water molecules. A similar observation was also made in the *S. thermophilus* structure[13], providing another piece of evidence in support of our speculation.

**Asymmetric interactions with RuvA**

To understand the mechanisms of RuvB recruitment to the HJ DNA by RuvA, we reconstituted the complex of RuvA-RuvB-HJ DNA for structural analysis (Supplementary Fig. 10a). To our surprise, we only obtained a structure of RuvA-HJ DNA (Supplementary Fig. 10b–e, Supplementary Table 1). Perhaps, the entire complex readily dissociated during grid preparation. Cryo-EM structure of the RuvA-HJ DNA complex revealed that RuvA assembles as a tetramer to recognize the crossover of HJ DNA through charge-charge interactions (Supplementary Fig. 10d, e), which is consistent with previous biochemical and structural analysis[10,13].

Previous studies have revealed that RuvB C-terminal domain interacts with RuvA with a stoichiometry of 6:4[10,13,27]. The asymmetric assembly of RuvB hexamer provides a rational explanation for the previously mentioned studies[10,27]. To dissect which protomers in the

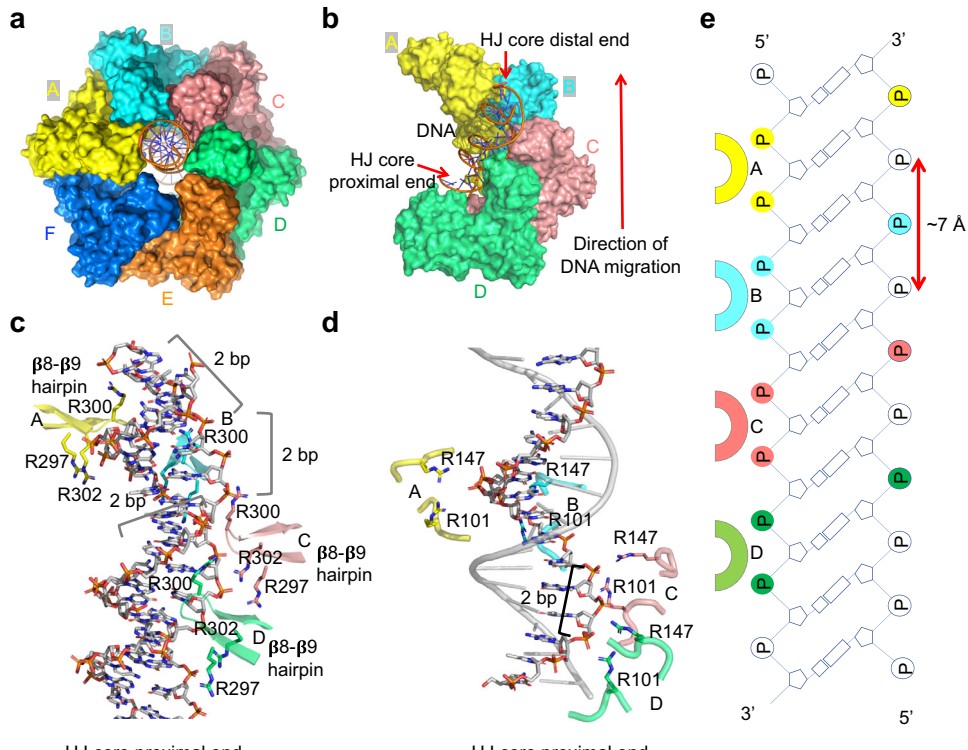

**Fig. 4 | DNA engagement by RuvB. a** Surface representation of RuvB-DNA complex with dsDNA highlighted as orange sticks and subunits colored as A (yellow), B (cyan), C (salmon), D (green), E (orange), and F (blue). The same color scheme is used throughout the figure. **b** DNA in complex with the four DNA-engaging RuvB protomers, revealing an architecture of a spiral staircase by protomers A (yellow), B (cyan), C (salmon), and D (green). The direction of DNA migration was indicated by an upward red arrow. **c** Ribbon diagrams showing interactions between dsDNA and the β8-β9 hairpin (Arginine finger) in the four DNA-engaging subunits. From one subunit to another, a RuvB protomer rotates about 60 degrees and rises 2 bp per revolution step, following along the DNA helix. **d** Ribbon diagrams showing interactions between dsDNA and Arg101 and Arg147 in the four DNA-engaging subunits. **e** Schematic of the interface between the dsDNA substrate and the four staircase RuvB protomers.

RuvB hexamer can interact with RuvA, we docked the RuvA CTD to the RuvB NTD based on the crystal structure of the RuvB and RuvA CTD complex, which reveals that the feature β-hairpin in RuvB NTD is critical for the recruitment of RuvA CTD[10]. Protomers A, F, E, and D have sufficient space to establish interactions with RuvA (Fig. 6a). In contrast, protomers B and C cannot accommodate the CTD of RuvA due to clashing of residues induced by steric hindrance with neighboring RuvB protomers (Fig. 6b, c). Therefore, the asymmetric assembly of the RuvB hexamer determines the 6:4 stoichiometry between RuvB and RuvA in the HJ complex. As RuvA can only engage with four protomers, we speculate that RuvA domain III undergoes cycles of transient dissociation from and reattachment to RuvB during HJ branch migration at the A, F, E, and D positions, similar to a carbon brush in an electric generator. Furthermore, the intrinsically disordered nature of the feature β-hairpin in RuvB may also facilitate the dynamic interactions between RuvA and RuvB.

**Mechanisms of Holliday junction migration**
Our structural analysis allowed us to propose a model for the HJ branch migration catalyzed by the asymmetric RuvB hexamer (Fig. 6d, Supplementary Movie 1). RuvB protomers are recruited by RuvA to the HJ and assemble into an asymmetric hexamer bound to ATP. The sequential cycling of ATP hydrolysis, ADP release, and ATP exchange triggers concerted conformational changes of RuvB protomers, leading to the revolution of RuvB around duplex DNA, accompanied by a pulling motion of DNA from the HJ core (Fig. 6d). HJ branch migration is advanced by two nucleotides at the cost of one ATP molecule for each step or 12 nucleotides at the cost of 6 ATP in one complete revolution of RuvB. Due to steric hindrance in positions B and C, RuvA

undergoes cycles of dissociation and reattachment to RuvB protomers, which also prevents dsDNA from forming a knot, kinking, breaking, or building up excessive torsion during HJ migration.

## Discussion
In earlier studies, two models have been proposed to explain the molecular mechanisms of AAA+ motor proteins: one being the rotation model and the other a revolution model[28]. In the rotation model, motor proteins rotate on their own axis, while revolution motor proteins revolve around their substrates, similar to a planet's day vs year cycle. Structurally, rotation motors tend to form oligomers with a small central pore (diameter <2.0 nm), and revolution motors generally display a large central pore (diameter >3.0 nm)[28]. All protein translocases with structures available for analysis, including Cdc48, p97, and Clpx, display a small pore and have five subunits engaged with their substrates[17–20]. Our structure of RuvB hexamer clearly showed that RuvB forms a ring-like structure with a large central pore (3.4 nm), which has enough space for RuvB to revolve around the DNA duplex, and that RuvB has four protomers engaging DNA, contrasting those with small pores.

It may be a universal mechanism that AAA+ ATPases tend to form asymmetric spiral staircases in the presence of substrates for function. Earlier AAA+ ATPase studies revealed that most AAA+ ATPases form symmetric oligomers[14,15,24,29–32]. At that time, most structures of AAA+ ATPase were determined in the absence of substrates due to technical challenges in obtaining crystals of AAA+ ATPases in complex with substrates. With the advancement of cryo-EM single particle analysis, more and more cryo-EM structures of AAA+ ATPase in complex with substrates were published, revealing that a spiral assembly of AAA+

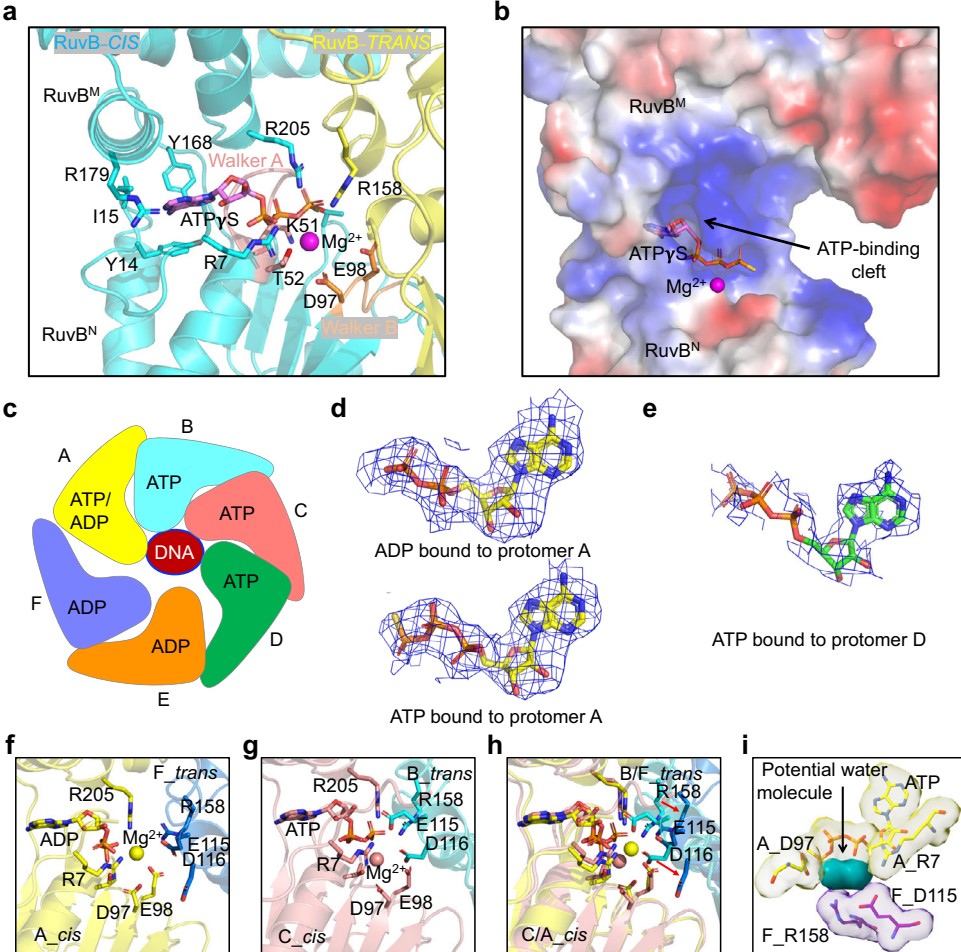

**Fig. 5 | Nucleotide-binding sites in RuvB hexamer. a** ATP-binding site in a representative RuvB dimer. ATP is coordinated by two protomers of RuvB (*cis* in blue and *trans* in yellow). ATP and residues responsible for coordinating ATP are highlighted as sticks. The magnesium ion for catalysis was highlighted as a sphere (pink). Walker A motif and Walker B motif were colored in salmon and orange, respectively. **b** Electrostatic surface representation of the RuvB *Cis* subunit showing the binding pocket of ATP with positively charged regions in blue and negative charge in red. **c** Diagram illustrating the bound nucleotides in the six protomers of RuvB hexamer with A (yellow), B (cyan), C (salmon), D (green), E (orange), F (purple), and DNA (red). **d** ADP or ATP fitted to the cryo-EM density maps (blue

mesh) in protomer A at 2.5 σ or 1.8 σ, respectively. **e** ATP fitted to the cryo-EM density maps (blue mesh) in protomer D at 1.8 σ. **f** The ATP catalytic center in the AF dimer with key residues for catalysis highlighted as sticks. Protomers A and F are shown in yellow and blue, respectively. **g** The ATP catalytic center in the CB dimer with key residues for catalysis highlighted as sticks. Protomers C and B are shown in salmon and cyan, respectively. **h** A comparison of the ATP catalytic centers in the AF dimer and the CB dimer reveals dramatic conformational changes in the trans protomers. The color schemes match (**f**, **g**). **i** Cryo-EM density maps (2.0 σ) of the ATP catalytic center in the A (yellow) and F (purple) dimer, showing the unmodelled density (green) as potential water molecules.

ATPase hexameric rings is universally shared to facilitate the translocation of substrates through the pore. In our case, we also observed that RuvB assembles into a spiral staircase with dsDNA in the central pore. More interestingly, four protomers of RuvB hexamer engage with dsDNA, indicating a coupled mechanism of revolution and translocation by RuvB to facilitate Holliday junction migration. In addition, the dsDNA in our structure is longer than what was observed in the recently published HJ complex structure[13], which allowed us to identify two DNA binding motifs in a RuvB protomer.

Furthermore, ATP binding, ATP hydrolysis, and ADP release are key events driving the conformational changes of AAA+ ATPases for triggering the "hand-over-hand" movement of the arginine finger pore loops that facilitate the unidirectional movement of substrates. At least three models have been proposed for mechanisms of ATP hydrolysis by AAA+ ATPases, including a synchronized model, stochastic model, and sequential model[33–35]. Our structural study on RuvB, together with a previous structural study on *S. thermophilus* RuvB, came to a census that RuvB hydrolyzes ATP most probably in a sequential manner, in which ATP hydrolysis occurs at the uppermost position of the spiral staircase, while ATP is loaded at the lowest point of the spiral staircase.

In the published manuscript on *S. thermophilus* RuvB, it was proposed that ADP release triggers ATP hydrolysis through a signaling mechanism across protomers of RuvB hexamer[13]. However, from our structural analysis of RuvB hexamer, we did not see strong evidence to support this cross-protomer communication. Instead, we tend to believe that ATP hydrolysis occurs at the top position of the spiral staircase mainly because the RuvB protomer at that point and its neighboring protomer in *trans* is better positioned for ATP hydrolysis than the other subunits. In addition, the relatively poor density of ATPγS in the nucleotide-binding pocket of protomer D suggested that ATPγS may have relatively low occupancy. Some particles may have ADP or are empty at the position of protomer D. Considering that protomer E is occupied by ADP while protomer C has an ATP in its pocket, we posit that the exchange of ADP to ATP takes place at the position of protomer D, consistent with what was proposed in a recent publication[13].

Lastly, RuvB is loaded onto HJ DNA by interacting with RuvA, which assembles as a tetramer to bind to the HJ core (Supplementary Fig. 10d, e). The asymmetry of RuvB's hexameric assembly provides a mechanistic explanation for the 6:4 stoichiometry between RuvB and RuvA, which is

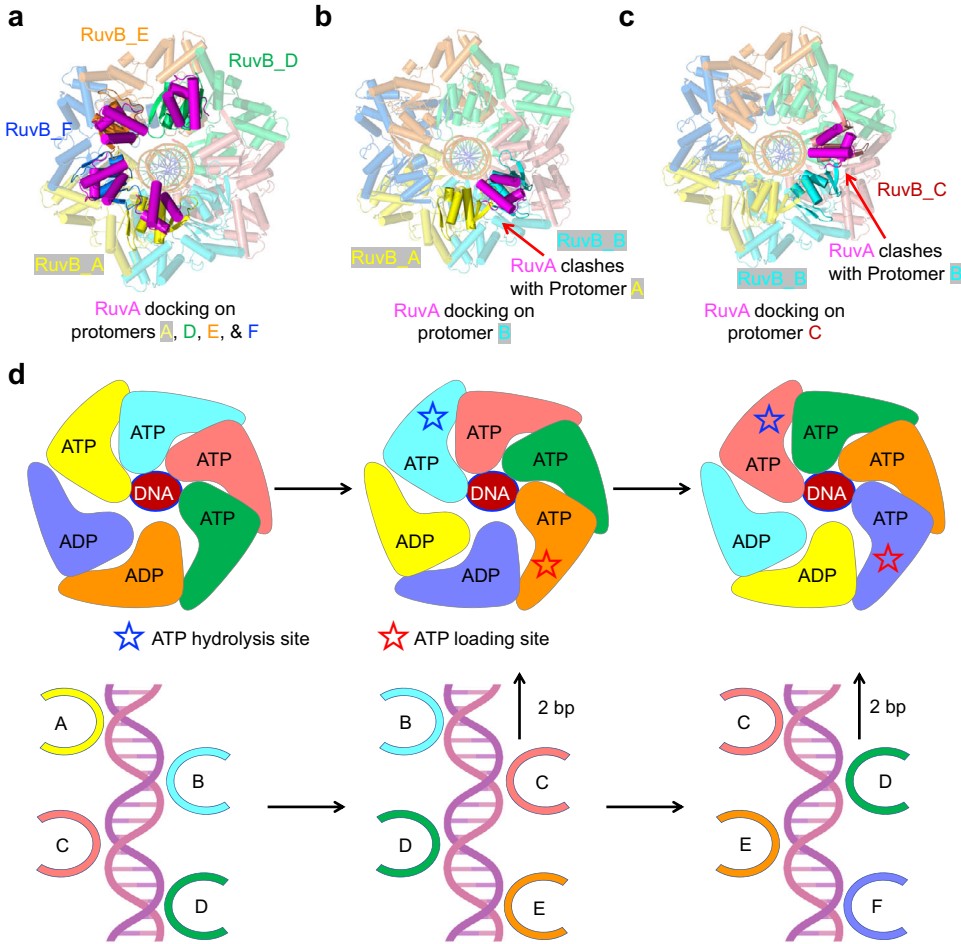

**Fig. 6 | Mechanisms of RuvA engagement and Holliday junction migration.**
**a** Ribbon diagram showing the interactions between RuvA (magenta) and RuvB protomers A (yellow), D (green), E (orange), and F (blue) using a docked model.
**b** Ribbon diagram showing the interactions between RuvA (magenta) and RuvB protomer B (cyan), revealing that RuvA at position B can clash with protomer A (yellow). **c** Ribbon diagram showing the interactions between RuvA (magenta) and RuvB protomer C (salmon), revealing that RuvA at position C can clash with protomer B (cyan). **d** Diagram illustrating a mechanism of Holliday junction migration. RuvB and DNA assemble into a spiral staircase with four protomers engaged with the DNA substrate. The ATP was hydrolyzed in a sequential manner and occurred one at a time at position A (blue star). At position D, ADP was exchanged for ATP (red star). ATP hydrolysis drives the conformational changes of RuvB accompanied by revolving around and pulling of dsDNA by two base pairs at a time. The color scheme is the same as in **a**.

also supported by the complex structure of RuvA/RuvB/HJ published recently[13]. Domain III of RuvA, which is linked to the CTD RuvB winged helix DNA binding domain through a flexible linker, forms interactions with the feature β-hairpin of RuvB, which is a conformationally flexible motif in RuvB. These features enable a mechanism of association and detachment between RuvA and RuvB during the revolution of RuvB, similar to the working mechanism of the carbon brush and the motor in an electric generator. Thus, the concerted events of nucleotide cycling, RuvB conformational oscillation, DNA pulling, and rotation, as well as RuvA association and detachment, ensure a precise DNA migration by a step size of 2 nucleotides per ATP hydrolyzed.

## Methods
### Protein expression and purification
RuvB and RuvA from *T. thermophilus* were cloned into a pET-28a(+) expression vector with an N-terminal histidine tag, using the NdeI/SalI and NdeI/XhoI restriction sites for RuvB and RuvA, respectively, using the following primers: RuvA_F (GGAATTCCATATGATCC GCTACCTCC GGGGCCTGG), RuvA_R (CCGCTCGAGCTAGCGGAGGCGCTTTAGG GC CTCC), RuvB_F (GGAATTCCATATGGAAGACCTCGCCCTTAGGCCC), and RuvB_R (ACGCGTCGACTCACGGCTCCAAGAGGGGGCCCACC). RuvB and RuvA were recombinantly expressed using *E. coli* BL21(DE3)

RipL strain. Bacterial cells were grown at 37 °C in LB supplemented with 50 µg mL$^{-1}$ kanamycin until they reached an OD$_{600}$ of about 0.6. Expression of RuvB or RuvA was induced by the addition of 0.5 mM isopropyl β-D-1-thiogalactopyranoside (IPTG), and cultures were further incubated at 18 °C for 16 h. Cells were pelleted at 2000 × *g* for 20 min at 4 °C, resuspended in 20 mM Tris-HCl pH 7.5, 300 mM NaCl lysis buffer, supplemented with a complete mini protease inhibitor cocktail (Roche, SKU 11836153001), disrupted by sonication, incubated in a water bath at 55 °C for 30 min, and the resulting lysate was centrifuged at 40,000 × *g* for 60 min at 4 °C. The supernatant was loaded onto a column containing Ni-NTA beads (Qiagen, catalog no. 30210) that were pre-equilibrated with lysis buffer. The column was washed with 10 column volumes (CV) of 20 mM Tris-HCl pH 7.5, 300 mM NaCl, 30 mM imidazole wash buffer, followed by elution of RuvB using 5 CV of 20 mM Tris-HCl pH 7.5, 300 mM NaCl, 300 mM imidazole elution buffer, in 1 mL aliquots. Eluted protein was pooled, dialyzed against 20 mM Tris-HCl pH 7.5, 150 mM NaCl gel filtration buffer, and concentrated to 2 mL using an Amicon ®Ultra 30,000 NMWL centrifugal filter (Millipore, SKU UFC903024), before application to a Superdex 200 10/300 GL Increased column (Cytvia, GE28-9909-44) to purify protein to homogeneity. RuvB was eluted with gel filtration buffer on an ÄKTA go FPLC (Cytvia). RuvB and RuvA were verified by SDS-Page gel, stained with

Coomassie blue, and protein concentration was determined by nanodrop A280 (35,973 MW, extinction coefficient 11,920).

### RuvB/DNA complex and RuvA-RuvB-HJ DNA preparation

The HJ DNA substrate was prepared by annealing the four following synthetic DNA oligonucleotides[25]: HJ1 (AGAATCTGCCGAGAGACCG AGCAGAATTCTATGTGTTT ACCAAGCGCTG), HJ2 (CAGCGCTTGGTAA ACACATAGAATTCTGCTCGGTCTGAGC CGTCTAAGA), HJ3 (TCTTA-GACGGCTCACTGGCTGTGGGATCCGAGCTGTCTAGAG ACATCGA), HJ4 (TCGATGTCTCTAGACAGCTCGGATCCCACAGCCAGTCTCGGCA GATTCT). The four strands were mixed at equimolar ratios and were incubated at 95 °C for 10 min before allowing to cool to room temperature. The RuvB/DNA complex was reconstituted by mixing the purified RuvB with synthesized HJ DNA at 1:6 (DNA/RuvB monomer ratio) in 20 mM Tris-HCl pH 7.5, 10 mM MgCl$_2$, 1 mM DTT, and 2 mM ATP-γ-S buffer for 30 min at 37 °C. The complex was purified from monomeric RuvB via gel filtration using 20 mM Tris-HCl, 150 mM NaCl buffer. Complex purity was determined by SDS-Page gel, and DNA presence was determined by agarose gel stained with ethidium bromide. Only the peak pertaining to RuvB/DNA complex was used in further testing.

The entire complex of RuvA-RuvB-HJ DNA complex was prepared in the same conditions as the RuvB-DNA complex, using a 6:4:1 (RuvB:RuvA:HJ DNA) molar ratio.

### Data collection

Quantifoil 1.2/1.3, 300 mesh copper grids (Quantifoil) were glow discharged at 0.2 atm for 30 s and were loaded onto a FEI vitrobot (Thermo Fisher). 3 μL of the sample was applied to the grid face before blotting for 4 s and plunge freezing in liquid ethane. Grids were stored in liquid nitrogen prior to screening on a 200 kV Glacios (Thermo Fisher Scientific) equipped with a K3 bioquantum detector (Gatan, Inc.). Grids were again stored in liquid nitrogen until data could be collected using a 300 kV Titan Krios (Thermo Fisher Scientific), also equipped with a K3 detector. 6870 movies were collected, using data acquisition software EPU (v2.12.1.2782REL), and TIA (v5.0 SP4) (Thermo Fisher Scientific), at a magnification of ×81,000 and a pixel size of 1.08 Å, at a total dose of 50 e$^-$/Å$^2$. All data processing, including patch motion correction and contrast transfer function (CTF) calculations, were performed in cryoSPARC (ver. 3.2)[36].

### Structural reconstruction and model building

Approximately 500 particles were manually picked from 65 exposures for initial 2D classification, creating six 2D classes that were used to train the template picker job. After manual inspection of template picker picks, 6,874,881 particles were picked from 6723 micrographs that showed clear features of target particles. Two rounds of 2D classification were performed to remove poor-quality particles. Multi-class ab-Initio reconstructions were performed using 884,750 particles and yielded four volumes for subsequent 3D classification ranging from 3.51-6.59 Å, estimated by the "gold standard" Fourier shell correlation of 0.143[37]. Heterogenous refinement of an Ab-initio volume containing 305,449 particles followed by non-uniform refinement produced a final volume at 3.2 Å, using C2 symmetry. Additionally, a mask was generated over a map volume pertaining to a single hexamer of the 3.2 Å map in UCSF Chimera (ver. 1.15), which was used in conjunction with the local refinement of a heterogeneously refined volume containing 390,143 particles to yield a final volume at 2.97 Å using C1 symmetry. The two final volume maps were used for model building, post-processing, and interpretation using Phenix (ver. 1.20.1), UCSF Chimera (ver. 1.15), chimeraX (v1.4 and v1.5), coot (ver. 0.9.8.1), ISOLDE v1.5, Molprobity (unversioned), and PyMol (ver. 2.1)[38–40]. Coordinates of the RuvB dodecamer and RuvB hexamer atomic models, 8EFY and 8EFV, respectively, have been deposited to the Protein Data Bank, and their density maps, EMD-28107 and EMD-28101 have been deposited to the Electron Microscopy Data Bank.

Similar procedures were employed for the reconstruction of RuvA-HJ DNA. 3,383,678 particles were initially picked, and after 2D classification 130,426 particles were used for 3D reconstruction. The volume was refined with a C4 symmetry and a final map at 4.3 Å resolution was obtained. Coordinates of the RuvA-HJ DNA atomic model, 8GH8, have been deposited to the Protein Data Bank, and the density map, EMD-40036, has been deposited to the Electron Microscopy Data Bank.

### Reporting summary

Further information on research design is available in the Nature Portfolio Reporting Summary linked to this article.

## Data availability

Accession numbers for RuvB dodecamer, RuvB hexamer, and RuvA-HJ DNA are as follows: (coordinates of atomic models: 8EFY, 8EFV, and 8GH8 deposited to Protein Data Bank), and (density map: EMD-28107, EMD-28101, and EMD-40036 deposited to Electron Microscopy Data Bank). Additional atomic coordinate models used in this study including 1HQC, 7PBS, 7PBU, and 2HOI are available in the Protein Data Bank. All data needed to evaluate the conclusions in the paper are presented in the paper.

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

## Acknowledgements
We thank Xiaoyuan Yang for her assistance in making figures. Grid screenings were performed at OSU CEMAS with the assistance of Drs. Giovanna Grandinetti and Yoshie Narui. Cryo-EM data for RuvB-DNA were collected with the assistance of Drs. Adam D. Wier, Thomas J. Edwards, Tara Fox, and Jenny Wang at the National Cancer Institute Cryo-Electron Microscopy Center supported by grants from the NIH National Institute of General Medical Sciences (GM103310). Cryo-EM data for RuvA-HJ were collected with the assistance of Dr. Patrick James Pascual at Stanford-SLAC cryo-Electron microscopy center. A.D.R. and T.M.F. are supported by an NIH T32 (GM118291-05 and GM144293- 01) and NIH R35 (GM1R35GM147465-01), respectively.

## Author contributions
T.M.F. conceived the project. A.D.R., Z.C., and T.M.F. performed molecular cloning of and biochemical purification of RuvA and RuvB, and the biochemical reconstitution of the RuvB-DNA complex. Z.S., T.M.F, and A.D.R. prepared grids, determined the cryo-EM structures, and built the models. N.Z. and Q.Z. did mass spectrometry analysis. T.M.F., A.D.R., and Z.S. analyzed the data. T.M.F. and A.D.R. wrote the manuscript with inputs from all the authors.

## Competing interests
The authors declare no competing interests.
