## [Peer Review File · Nature Communications]

Molecular mechanisms of Holliday junction branch migration catalyzed by an asymmetric RuvB hexamerREVIEWER COMMENTS

Reviewer #1 (Remarks to the Author):

Fu and colleagues presented the cryo-EM structures of the *S. thermophilus* AAA+ ATPase protein RuvB in complex with DNA and adenine nucleotides. RuvB is known to drive the branch migration of the Holliday junction universal in DNA homologous recombination. The authors discovered that six RuvB subunits arrange themselves into a helical-shaped split ring, resembling a split lock washer. DNA is enclosed in the central hole. Among the six subunits, only four contact the DNA backbone. Each of the four subunit contacts exactly two nucleotides (and one of their complementary nucleotides on the opposite strand), suggesting a DNA translocate movement of 2-nucleotide step. Moreover, the subunits show different adenine nucleotide binding state, providing a possible mechanism for coupling ATP hydrolysis/ADP-ATP exchange with DNA translocation. Lastly, the asymmetric nature of the RuvB hexamer also explains the 6:4 stoichiometry between RuvB and its binding partner RuvA. The structural analyses are extensive and thorough but certain reasonings can certainly be improved.

The reviewer has the concerns listed below.

-- From the Method section, the authors used Holliday Junction DNA to obtain the RuvB-DNA complex, yet in both dodecamer and hexamer RuvB-DNA structures, the DNA molecules are double-stranded. From the structures, is it possible to deduce the direction of migration of DNA (relative to RuvB)? This is particularly intriguing as the RuvB hexamer is asymmetric and the subunits are different in terms of DNA interaction and ATP/ADP binding status. The author observed that subunit B, C, and D in the hexamer bind ATP while A, F, and E bind ADP, though subunit A may bind to either ATP or ADP. Based on this observation, the authors proposed that ATP hydrolysis takes place between subunit F and A, "at the top of the staircase". However, the residues coordinate ATP hydrolysis are not well positioned in FA dimer. Can the authors distinguish their model from the alternative one that FA dimer represents an ADP-ATP exchange conformation, while ATP hydrolysis takes place in DE dimer?

-- The authors prepared their RuvB-DNA complexes using 2 mM ATPγS. Is monomeric RuvB nucleotide-free or bound with ADP? Is ATPγS essential for the assembly of the RuvB-DNA complex? Later they reported the RuvB hexamer and dodecamer contain ATP and ADP. Are the resolutions sufficient to tell ATP from ATPγS? Could poor densities lead to some ATPγS mistakenly being interpreted as ADP? Since 2 mM ATPγS is used for complex assembly, it is difficult to envision the ADP molecules are co-purified from bacterial culture. ATPγS is known to slowly hydrolyze, was the incubation condition amenable for ATPγS hydrolysis to yield ADP molecules? Clarification of these points will help to define the conformations of the RuvB subunits, e.g., post- or pre-hydrolysis.

-- In Figure 1A and 1C the two hexamers in the dodecamer RuvB-DNA complex seem to weakly interact with each other. Is this interaction mediated by the feature β hairpins? Is this RuvB-RuvB interaction biological relevant? The author may design mutation or cite literature to address the issue.

-- Page 6, "The relative orientations of RuvB^C, RuvB^M, and RuvB^N display clear deviations between protomers (Fig. 2C)." I could see clear swaying of the NTDs in subunits D and E (from A) but not B, C, or F. If this is due to structure orientation, please include a table to list the orientation changes in angles.

-- Page 6, "we propose that the assembly of RuvB-DNA complex may be a cooperative process involved in multiple sequential events, including RuvB-DNA binding, DNA engagement triggered RuvB conformational changes and the establishment of neighboring RuvB-RuvB interactions (Figure 2E)". This statement seems a huge logical leap to me. It is also vague and self-conflicting. The authors should clarify the statement by listing a few sequential steps, e.g., ATP binding, DNA binding, and state which process is the cooperative step, e.g. establishment of RuvB-RuvB interactions. Figure 2E middle panel "DNA-binding" not "DNA-biding".

-- Page 7, "This observation is consistent with the different functional roles of protomers A, B, C, and D during DNA translocation." What are the functional roles? If there are references, please cite them.

Other issues:

Both RuvB^N, RuvB^M, RuvB^C and NTD, MD, and CTD are used in the text and figures. Though they might be interchangeable, it will be easier for readers to follow if the authors use one naming system throughout.

Page 5: "RuvB assembles into a ring-like hexamer with dsDNA in the central pore that directly contacts four subunit (Fig. 1C-D)". It will help reading and understanding if the authors list the four subunits in text or label them in the figures.

Greek signs, β , some γ , and some letters in italic look different from the rest of the manuscript, though this may be a hiccup during PDF conversion.

Figure 1E-G, the feature β hairpins are very difficult to see due to their small sizes and positions. The authors may consider highlighting the hairpins or muting the rest of the molecules to increase contrast.

Fig. S1 Please add/mark protein standards to both the SEC profile and the SDS-PAGE gel.

Reviewer #2 (Remarks to the Author):

Rish et al describes the cryoEM structure of a prokaryotic RuvB hexamer bound to DNA and nucleotides. The work is well performed and the structure is interpreted to provide some clues about the mechanisms of Holliday junction (HJ) branch migration.

Major points

- One major issue is that the manuscript does not provide much new mechanistic insights about HJ branch migration, after comparison with the recent paper by Wald et al (Nature, 15 sept 2022) where the authors used time-resolved cryo-EM to characterize 5-7 distinct conformations of the RuvAB-HJ complex.

- A second important issue is that this work focuses on the RuvB-DNA complex, because the authors did not manage to obtain "useful datasets" for the HJ complex containing RuvB, RuvA and DNA.

Although it is in part correct, as mentioned by the authors, that this can be sufficient to reveal the mechanism of HJ branch migration, this argument suffers when there is a structure of the full RuvA-RuvB-DNA complex already published, and which represents better what happens during HJ branch migration.

- The authors incubate RuvB-DNA with ATP γ S, but they find ADP in the structure. I assume this is because ATP γ S, being a slow hydrolyzing analog, hydrolysis happens when preparing the grid, but the authors could clarify this specifically.

Minor points

- During image processing, it seems that C2 symmetry was used to obtain the map of dodecamers. If this is correct, why to impose a symmetry that might mask differences between the rings when this was probably not required to reach high resolution?

- The first time they use *T. thermophilus* or *E. coli*, better to use the full name

Reviewer #3 (Remarks to the Author):

In the manuscript presented by Rish et al., the authors describe the cryo-EM structure of a RuvB hexamer and dodecamer assembled on double-stranded DNA. The RuvB hexamer is a motor protein of the AAA+ ATPase family and collaborates with RuvA to resolve Holliday junctions in DNA. The cryo-EM map allows the authors to describe, or predict in some cases, the adenosine nucleotide state of each protomer in the assembled RuvB complex. The study also describes the molecular interactions of four of the six RuvB protomers with the double-stranded DNA the hexamer is assembled around. A key strength of the study is its contribution toward understanding the molecular mechanisms involving coordination of RuvB-DNA binding and oligomerization. The study further expands on the diverse roles of ATP binding and hydrolysis and how those events are associated with DNA interactions in the RuvB hexamer. That said, the structure-heavy study could be enhanced with some functional data to better understand the molecular mechanism of RuvB hexamers on different DNA substrates.

A major issue is that the RuvB hexamer is assembled on duplex DNA, as opposed to a Holliday junction. Yet the study does not include any discussion regarding the physiological relevance of this assembly. Since the DNA substrate in the structure is dsDNA as opposed to the intended Holliday junction arrangement, the authors need to address whether RuvB-dsDNA interactions/unwinding is relevant or whether the assembly is an artifact of the annealing/assembly process. Even if RuvB does not normally act on duplex DNA, an artificially assembled complex would still help to understand molecular interactions of protein and DNA and could be used for interpretation of the native cellular function. If this is the case, the authors should be transparent and straightforward in what they are interpreting and should be cautious not to lead the reader into assuming that duplex DNA and Holliday Junctions are the same.

Similar to the above, it is unclear whether RuvB is able to unwind duplex DNA. This point is especially true for the dodecamer model with the two hexamers facing one another on the same DNA strand. Some DNA unwinding assays and added clarification in the text would help.

Since a structure of the RuvA/RuvB/DNA complex could not be determined, the authors relied on modeling to try to predict RuvA interactions. However, some biochemical data would really help to define the contributions of RuvA in the context of the studies and help to resolve lingering questions. Is RuvA necessary for resolving dsDNA? Does RuvA enhance unwinding of duplex DNA by RuvB? Does the RuvB complex (with or without RuvA) resolve dsRNA (especially since the protein-nucleic acid are coordinated through the DNA backbone)?

The recent RuvA-RuvB structure communicated by Wald et al (ref 11 in the manuscript) should be further integrated into the context of the present work. The discussion section touches on the proposed mechanistic similarities and differences that can be concluded from the two structures, but the different arrangements of RuvB hexamers in the two structures and the unwinding of DNA in a HJ versus duplex DNA should be addressed.

Last section of intro states that the asymmetric assembly of RuvB indicates an asymmetric engagement with RuvA. However, this conclusion appears to be speculative as there is no RuvA being used in the present structural studies. Ideally, the authors should consider testing this hypothesis directly. As an alternative, it is suggested that the authors rephrase this statement to more accurately reflect the data being presented.

Minor:

The introduction highlights what is known about the RuvA-RuvB complex with a primary emphasis on its structure. It would be helpful if the authors could add to the intro a series of features/functions that are not known, despite those earlier structures, and describe how the present study helps to better understand HJR and the role of RuvA-RuvB in HJR.

It is unclear how protomers A, F, and E coordinate ADP molecules. Have the authors trapped a stable hexamer complex or is the arrangement of nucleotide more related to kinetics of unwinding and the nature of ATPyS (that is slowly hydrolyzed)? Some additional text should help to clarify the nature of how the complex is stalled in the state that it is.

"The RuvB hexamer has approximate dimensions of $80 \text{ \AA} \times 127 \text{ \AA} \times 127 \text{ \AA}$ (Fig. 1B & 1C)." should refer to (Fig. 1B & 1D).

Figure 2E presents a reasonable mechanism of RuvB hexamer assembly. However, with little data to support the model (at least in the present communication), it should be presented as a "proposed" mechanism of hexamer assembly.

Figure 2, panel D label has a typo. DNA-biding should be DNA-binding.

1st paragraph of "DNA Binding by RuvB" section states "at the end of the HJ core-distal end." This sentence should be edited since the complex is assembled without a HJ. Similarly, the last sentence of discussion could more accurately indicate that DNA (as opposed to HJ) is predicted to occur at 2 nts per step.

1st paragraph of "Nucleotide Cycling" has a typo at "the six promoters of RuvB hexamer": promoters should be protomers. Same with the next sentence "Promoters B, C, and D bind ATP....."

Response to Reviewers' Comments

Reviewer #1 (Remarks to the Author):

Fu and colleagues presented the cryo-EM structures of the *S. thermophilus* AAA+ ATPase protein RuvB in complex with DNA and adenine nucleotides. RuvB is known to drive the branch migration of the Holliday junction universal in DNA homologous recombination. The authors discovered that six RuvB subunits arrange themselves into a helical-shaped split ring, resembling a split lock washer. DNA is enclosed in the central hole. Among the six subunits, only four contact the DNA backbone. Each of the four subunit contacts exactly two nucleotides (and one of their complementary nucleotides on the opposite strand), suggesting a DNA translocate movement of 2-nucleotide step. Moreover, the subunits show different adenine nucleotide binding state, providing a possible mechanism for coupling ATP hydrolysis/ADP-ATP exchange with DNA translocation. Lastly, the asymmetric nature of the RuvB hexamer also explains the 6:4 stoichiometry between RuvB and its binding partner RuvA. The structural analyses are extensive and thorough but certain reasonings can certainly be improved. We thank the very positive comments of reviewer 1. In this revised version, we have used native mass spectrometry to reveal that RuvB binds to endogenous ADP. We also reconstituted the entire RuvA-RuvB-HJ DNA complex and determined the cryo-EM structure of RuvA-HJ. We also revised our manuscript thoroughly based on the reviewer's comments.

The reviewer has the concerns listed below.

-- From the Method section, the authors used Holliday Junction DNA to obtain the RuvB-DNA complex, yet in both dodecamer and hexamer RuvB-DNA structures, the DNA molecules are double-stranded. From the structures, is it possible to deduce the direction of migration of DNA (relative to RuvB)? This is particularly intriguing as the RuvB hexamer is asymmetric and the subunits are different in terms of DNA interaction and ATP/ADP binding status.

Yes, we can deduce the direction of DNA migration. Based on structural analysis, RuvB protomers rotate in the order of A-B-C-D-E-F and the DNA migrates from the NTD of RuvB towards the CTD of RuvB as illustrated in our supplemental movie (Also see below). The conclusion from our structural analysis is consistent with previous biochemical studies^{1,2}. We have revised our Fig. 4B by adding an arrow to illustrate the direction of DNA migration.

The author observed that subunit B, C, and D in the hexamer bind ATP while A, F, and E bind ADP, though subunit A may bind to either ATP or ADP. Based on this observation, the authors proposed that ATP hydrolysis takes place between subunit F and A, “at the top of the staircase”. However, the residues coordinate ATP hydrolysis are not well positioned in FA dimer. Can the authors distinguish their model from the alternative one that FA dimer represents an ADP-ATP exchange conformation, while ATP hydrolysis takes place in DE dimer?

From the cryo-EM density of our structure, subunit A binds both ATP and ADP with ADP as the dominant molecule. As ADP dominates the nucleotide-binding pocket, the FA dimer in our structure most probably represents an ATP post-hydrolysis state. In contrast, the cryo-EM density of nucleotides in subunit D is relatively weak and can be nicely fitted with ATP, suggesting that the nucleotide-binding pocket of subunit D is either empty or occupied by ATP. Collectively, we are confident to propose that FA dimer represents ATP hydrolysis point while ADP-ATP exchange occurs at subunit D. Our conclusion was supported by the recent structures of RuvB-HJ complex published in Nature.

-- The authors prepared their RuvB-DNA complexes using 2 mM ATP γ S. Is monomeric RuvB nucleotide-free or bound with ADP? Is ATP γ S essential for the assembly of the RuvB-DNA complex? Later they reported the RuvB hexamer and dodecamer contain ATP and ADP. Are the resolutions sufficient to tell ATP from ATP γ S? Could poor densities lead to some ATP γ S mistakenly being interpreted as ADP? Since 2 mM ATP γ S is used for complex assembly, it is difficult to envision the ADP molecules are co-purified from bacterial culture. ATP γ S is known to slowly hydrolyze, was the incubation condition amenable for ATP γ S hydrolysis to yield ADP molecules? Clarification of these points will help to define the conformations of the RuvB subunits, e.g., post- or pre-hydrolysis.

This is a great question. To address this question, we used native mass spectrometry to analyze our purified RuvB. We are very surprised to find that the majority of our purified

RuvB binds to ADP (Figure S9D, see below). Previous biochemical assays showed that ATP γ S facilitates the assembly of RuvB-DNA complex³⁻⁶.

Our resolution is not sufficient to tell ATP from ATP γ S but is sufficient to distinguish ATP γ S from ADP. Considering that we supplemented with ATP γ S and our apo RuvB mainly binds to ADP, we are confident that what we observed in our structure is ATP γ S, not ATP.

As the reviewers pointed out that ATP γ S can slowly hydrolyze, some of the ADP molecules may come from ATP γ S hydrolysis. Given our apo RuvB has ADP molecules, we believe that most ADP molecules we observed are endogenously bound to RuvB.

We have clarified these points in our revised manuscript (see below).

“To rule out the ADP molecules in protomers E and F are endogenous or generated by ATP γ S hydrolysis, we did native mass spectrometry analysis of our purified RuvB and found that the majority of our purified RuvB bind endogenous ADP molecules (Fig. S9D). Thus, the ADP molecules in protomers E and F were not exchanged into ATP γ S during our assembly of RuvB-DNA complex, suggesting that RuvBs at positions E and F prefer to bind ADP over ATP γ S.”

-- In Figure 1A and 1C the two hexamers in the dodecamer RuvB-DNA complex seem to weakly interact with each other. Is this interaction mediated by the feature β hairpins? Is this RuvB-RuvB interaction biological relevant? The author may design mutation or cite literature to address the issue.

Yes, the interaction is mediated by the feature β hairpins. Under physiological conditions, the feature β hairpins are responsible to interact with RuvA. Therefore, the dodecamer RuvB-RuvB interaction is not biologically relevant. We have revised our manuscript and made this point clear by adding a few sentences as below.

“Under physiological conditions, RuvB hexamer flanks RuvA-HJ DNA to catalyze the migration of HJ. As such, the RuvB dodecamer we obtained is not biologically relevant.”

-- Page 6, "The relative orientations of RuvBC, RuvBM, and RuvBN display clear deviations between protomers (Fig. 2C)." I could see clear swaying of the NTDs in subunits D and E (from A) but not B, C, or F. If this is due to structure orientation, please include a table to list the orientation changes in angles.

This is a great point. We have revised our figure 2 to make it more obvious and add the rotational angles on top of the structures in Fig 2C (see below).

-- Page 6, "we propose that the assembly of RuvB-DNA complex may be a cooperative process involved in multiple sequential events, including RuvB-DNA binding, DNA engagement triggered RuvB conformational changes and the establishment of neighboring RuvB-RuvB interactions (Figure 2E)". This statement seems a huge logical leap to me. It is also vague and self-conflicting. The authors should clarify the statement by listing a few sequential steps, e.g., ATP binding, DNA binding, and state which process is the cooperative step, e.g. establishment of RuvB-RuvB interactions. Figure 2E middle panel "DNA-binding" not "DNA-biding".

We agree with the reviewer's comments and have revised our manuscript as below:

"We propose that the assembly of RuvB-DNA complex may be a hierarchical process involved in multiple sequential events, in which one protomer of RuvB binds to DNA followed by conformational changes of RuvB and the cooperative assembly of additional RuvB-RuvB interactions."

-- Page 7, "This observation is consistent with the different functional roles of protomers A, B, C, and D during DNA translocation." What are the functional roles? If there are references, please cite them.

ATP hydrolysis occurs at protomer A while ATP loading takes place at protomer D, which was elaborated in next section. To make it clear, we have deleted this statement in our revised manuscript.

Other issues:

Both RuvBN, RuvBM, RuvBC and NTD, MD, and CTD are used in the text and figures. Though they might be interchangeable, it will be easier for readers to follow if the authors use one naming system throughout.

We thank the reviewer for their advice and have revised our manuscript accordingly.

Page 5: “RuvB assembles into a ring-like hexamer with dsDNA in the central pore that directly contacts four subunit (Fig. 1C-D)”. It will help reading and understanding if the authors list the four subunits in text or label them in the figures.

Greek signs, β , some γ , and some letters in italic look different from the rest of the manuscript, though this may be a hiccup during PDF conversion.

We thank the reviewer and have revised our figure 1D.

Figure 1E-G, the feature β hairpins are very difficult to see due to their small sizes and positions. The authors may consider highlighting the hairpins or muting the rest of the molecules to increase contrast.

We thank the reviewer and have revised our figures 1E-G as below.

Fig. S1 Please add/mark protein standards to both the SEC profile and the SDS-PAGE gel.

We have added protein standards in our revised figure S1 (See below).

Reviewer #2 (Remarks to the Author):

Rish et al describes the cryoEM structure of a prokaryotic RuvB hexamer bound to DNA and nucleotides. The work is well performed and the structure is interpreted to provide some clues about the mechanisms of Holliday junction (HJ) branch migration.

We thank the reviewer for their very positive comments. In this revised version, we have used native mass spectrometry to reveal that RuvB binds to endogenous ADP. We also reconstituted the entire RuvA-RuvB-HJ DNA complex and determined the cryo-EM structure of RuvA-HJ. We also revised our manuscript thoroughly based on the reviewers' comments.

Major points

- One major issue is that the manuscript does not provide much new mechanistic insights about HJ branch migration, after comparison with the recent paper by Wald et al (Nature, 15 sept 2022) where the authors used time-resolved cryo-EM to characterize 5-7 distinct conformations of the RuvAB-HJ complex.

We agree with the reviewer that the recent paper by Wald et al in Nature provides comprehensive insights by characterizing 5-7 distinct conformations of the RuvAB-HJ complex. Our overall conclusions align well with that paper. However, compared to that paper, our overall resolution of RuvB has a higher resolution with a much longer dsDNA substrate. Therefore, we were able to identify a second DNA-binding site in RuvB (Fig. 4D), which was not observed by the Nature paper. Furthermore, we did more rigorous work in analyzing the structures. For example, we elaborated on the conformational differences of each protomer and proposed a hierarchical assembly model of the RuvB-DNA complex (Fig. 2). We also compared the interfaces of neighboring RuvB and highlighted the asymmetric assembly of RuvB (Fig. 3). As such, we believe that our structures complement with the Nature paper and provide deep insights into the Holliday junction migration catalyzed by the asymmetric RuvB assembly.

- A second important issue is that this work focuses on the RuvB-DNA complex, because the authors did not manage to obtain "useful datasets" for the HJ complex

containing RuvB, RuvA and DNA. Although it is in part correct, as mentioned by the authors, that this can be sufficient to reveal the mechanism of HJ branch migration, this argument suffers when there is a structure of the full RuvA-RuvB-DNA complex already published, and which represents better what happens during HJ branch migration.

We agree that we did not obtain useful datasets for the entire RuvAB-HJ complex. In the HJ complex, the RuvA functions to recognize the HJ DNA and recruit RuvB. Crystal structures of RuvA-HJ DNA complex and monomeric RuvB-RuvA CTD have revealed all the mechanisms regarding the function of RuvA^{7,8}. Therefore, in this milestone structure of the entire RuvAB-HJ complex, most novel points actually come from the RuvB-DNA complex, which is also evidenced by the recent Nature paper focusing on the RuvB-DNA complex. As we have a structure of RuvB-DNA complex with better resolution and longer DNA substrate, our manuscript is comparable to the Nature paper and also provides additional information as elaborated above.

To obtain the entire complex of RuvAB-HJ complex, we tried different methods to prepare the complex and different ways to freeze the sample. Indeed, we could obtain a nice complex of RuvAB-HJ DNA on gel filtration (See below & Fig. S10A). The complex may fall apart during grid preparation. After we processed our data, we obtained a structure of RuvA with HJ. We incorporated this part in our revised manuscript (See below & Fig. S10B-E).

Fig. S10 Structure of RuvA-HJ DNA complex.

- The authors incubate RuvB-DNA with ATPγS, but they find ADP in the structure. I assume this is because ATPγS, being a slow hydrolyzing analog, hydrolysis happens when preparing the grid, but the authors could clarify this specifically.

This is a great question. As the reviewers pointed out that ATPγS can slowly hydrolyze, it is possible that ADP molecules may come from ATPγS hydrolysis. Another possibility is that ADP from *E. coli* is associated with purified RuvB. To address this question, we used native mass spectrometry to analyze our purified RuvB. We are very surprised to find that the majority of our purified RuvB binds to ADP (Figure S9D, see below also). As such, we tend to believe that the ADP in *E. coli* is associated with RuvB and was not replaced by ATPγS at specific positions of the hexamer though we cannot exclude the possibility that ADP comes from the hydrolysis of ATPγS. We have clarified this point in our manuscript in a few sentences as below.

“To rule out the ADP molecules in protomers E and F are endogenous or generated by ATP γ S hydrolysis, we did native mass spectrometry analysis of our purified RuvB and found that majority of our purified RuvB bind endogenous ADP molecules (Fig. S9D). Thus, the ADP molecules in protomers E and F were not exchanged into ATP γ S during our assembly of RuvB-DNA complex, suggesting that RuvBs at positions E and F prefer to bind ADP over ATP γ S.”

Minor points

- During image processing, it seems that C2 symmetry was used to obtain the map of dodecamers. If this is correct, why to impose a symmetry that might mask differences between the rings when this was probably not required to reach high resolution?

The dodecamer data was initially processed without imposing symmetry (Fig. S2). We compared the two hexamers and found that they aligned well with each other with minor differences in the NTD β -hairpin. As that part is not a major focus in the manuscript, we thus went ahead and applied C2 symmetry to improve our resolution.

- The first time they use *T. thermophilus* or *E. coli*, better to use the full name. We thank the reviewer to point out this and have corrected the names in our revised manuscript.

Reviewer #3 (Remarks to the Author):

In the manuscript presented by Rish et al., the authors describe the cryo-EM structure of a RuvB hexamer and dodecamer assembled on double-stranded DNA. The RuvB hexamer is a motor protein of the AAA+ ATPase family and collaborates with RuvA to resolve Holliday junctions in DNA. The cryo-EM map allows the authors to describe, or predict in some cases, the adenosine nucleotide state of each protomer in the assembled RuvB complex. The study also describes the molecular interactions of four of the six RuvB protomers with the double-stranded DNA the hexamer is assembled around. A key strength of the study is its contribution toward understanding the

molecular mechanisms involving coordination of RuvB-DNA binding and oligomerization. The study further expands on the diverse roles of ATP binding and hydrolysis and how those events are associated with DNA interactions in the RuvB hexamer. That said, the structure-heavy study could be enhanced with some functional data to better understand the molecular mechanism of RuvB hexamers on different DNA substrates.

We thank the reviewer for his/her invaluable comments. In this revised version, we have used native mass spectrometry to reveal that RuvB binds to endogenous ADP. We also reconstitute the entire complex of RuvA-RuvB-HJ DNA complex and determined the cryo-EM structure of RuvA-HJ. We also revised our manuscript thoroughly based on the reviewers' comments.

A major issue is that the RuvB hexamer is assembled on duplex DNA, as opposed to a Holliday junction. Yet the study does not include any discussion regarding the physiological relevance of this assembly. Since the DNA substrate in the structure is dsDNA as opposed to the intended Holliday junction arrangement, the authors need to address whether RuvB-dsDNA interactions/unwinding is relevant or whether the assembly is an artifact of the annealing/assembly process. Even if RuvB does not normally act on duplex DNA, an artificially assembled complex would still help to understand molecular interactions of protein and DNA and could be used for interpretation of the native cellular function. If this is the case, the authors should be transparent and straightforward in what they are interpreting and should be cautious not to lead the reader into assuming that duplex DNA and Holliday Junctions are the same. We thank the reviewer for the invaluable advice. Under physiological conditions, RuvB binds to the dsDNA part of the HJ DNA and functions to catalyze the migration of HJ. Therefore, our structures of RuvB in complex with dsDNA reflect the physiological process of HJ migration. Based on the reviewer's comments, we have revised our manuscript to avoid misleading our readers.

Similar to the above, it is unclear whether RuvB is able to unwind duplex DNA. This point is especially true for the dodecamer model with the two hexamers facing one another on the same DNA strand. Some DNA unwinding assays and added clarification in the text would help.

We apologized that we did not make this point clear. The function of RuvB is to hydrolyze ATP and promote the migration of dsDNA. Some earlier studies showed that RuvA-RuvB complex, but not RuvB alone, can unwind HJ DNA in the presence of ATP *in vitro*⁹. However, later studies revealed that RuvB can migrate along duplex DNA without unwinding dsDNA¹⁰. In any case, RuvB alone has not DNA helicase activity, which is consistent with our and others' structural analysis that RuvB assembles with a dsDNA substrate. We have revised our manuscript to clarify this point in the first paragraph of our introduction part (See below).

“As a motor protein, RuvB can bind the dsDNA part of the HJ and promote the movement of HJ, known as branch migration. Though earlier studies showed that RuvA-

RuvB complex, but not RuvB alone, can unwind dsDNA in the presence of ATP in vitro⁹, later studies revealed that RuvB drives branch migration by translocating along duplex DNA without unwinding dsDNA¹⁰.”

Since a structure of the RuvA/RuvB/DNA complex could not be determined, the authors relied on modeling to try to predict RuvA interactions. However, some biochemical data would really help to define the contributions of RuvA in the context of the studies and help to resolve lingering questions. Is RuvA necessary for resolving dsDNA? Does RuvA enhance unwinding of duplex DNA by RuvB? Does the RuvB complex (with or without RuvA) resolve dsDNA (especially since the protein-nucleic acid are coordinated through the DNA backbone)?

We apologized that we did not make this point clear. As we mentioned above, RuvA can recognize the structure of HJ DNA and mainly function to recruit RuvB onto HJ DNA. Though some earlier studies showed that RuvA can promote the DNA unwinding activity of RuvB in vitro, later evidence support that RuvB travels along dsDNA without unwinding dsDNA during Holliday junction migration. During our revision, we tried very hard to obtain the structure of the entire RuvA-RuvB-HJ DNA complex. We can obtain a nice complex containing RuvA, RuvB, and HJ DNA on gel filtration (See below and our added Fig.S10). However, when we determined the structure, we only saw RuvA-HJ DNA complex, which clearly revealed that RuvA can recognize the cross-point of HJ. The CTD of RuvA has been shown to directly bind and recruit RuvB. However, there is no evidence to show that this interaction regulates the activity of RuvB. From the structural analysis, RuvA binds to an inserted β -hairpin of RuvB NTD, which does not affect the ATP hydrolysis and DNA binding. As such, we believe that RuvA mainly functions to recruit RuvB to HJ DNA rather than affecting the HJ migration catalyzed by RuvB.

Fig. S10 Structure of RuvA-HJ DNA complex.

The recent RuvA-RuvB structure communicated by Wald et al (ref 11 in the manuscript) should be further integrated into the context of the present work. The discussion section touches on the proposed mechanistic similarities and differences that can be concluded from the two structures, but the different arrangements of RuvB hexamers in the two structures and the unwinding of DNA in a HJ versus duplex DNA should be addressed.

The overall assembly between our RuvB and the published RuvB in *Nature* is quite similar. Compared to the published one, the resolution of our structure is higher and the dsDNA in our structure is longer, which allows us to obtain more information about how

RuvB interacts with dsDNA. To better compare our structure with the published one, we have revised our manuscript and expanded our discussion on these points.

“In addition, the dsDNA in our structure is longer than what was observed in the recently published HJ complex structure, which allowed us to identify two DNA binding motifs in a RuvB protomer.”

Last section of intro states that the asymmetric assembly of RuvB indicates an asymmetric engagement with RuvA. However, this conclusion appears to be speculative as there is no RuvA being used in the present structural studies. Ideally, the authors should consider testing this hypothesis directly. As an alternative, it is suggested that the authors rephrase this statement to more accurately reflect the data being presented.

We thank the reviewer for the very generous comments. We indeed tried to assemble the entire complex during the revision (see above and Fig. S10). We can form the complex of RuvA-RuvB-DNA. However, the sample did not behave well on the grids. Unluckily, we failed to get a structure of the entire complex after many trials. We obtained the complex between RuvA and HJ DNA. As shown in Fig.S10, the RuvA binds to the core of HJ DNA. RuvB was supposed to bind to the dsDNA of HJ flanking RuvA. To better reflect the data presented, we added “potential” to tone down our statement in the revised manuscript based on the reviewers’ comments (see below).

“the asymmetric assembly of RuvB also indicates a potential asymmetric engagement with RuvA”

Minor:

The introduction highlights what is known about the RuvA-RuvB complex with a primary emphasis on its structure. It would be helpful if the authors could add to the intro a series of features/functions that are not known, despite those earlier structures, and describe how the present study helps to better understand HJR and the role of RuvA-RuvB in HJR.

This is a good point and we have revised our introduction based on the reviewer’s comment. Below are changes made in our revised introduction.

“As a motor protein, RuvB can bind the dsDNA part of the HJ and promote the movement of HJ, known as branch migration. Though earlier studies showed that RuvA-RuvB complex, but not RuvB alone, can unwind dsDNA in the presence of ATP in vitro⁹, later studies revealed that RuvB drives branch migration by translocating along duplex DNA without unwinding dsDNA¹⁰.”

“Additionally, the asymmetric assembly of RuvB also indicates a potential asymmetric engagement with RuvA.”

It is unclear how protomers A, F, and E coordinate ADP molecules. Have the authors trapped a stable hexamer complex or is the arrangement of nucleotide more related to

kinetics of unwinding and the nature of ATP γ S (that is slowly hydrolyzed)? Some additional text should help to clarify the nature of how the complex is stalled in the state that it is.

This is a great question. To reveal the origin of ADP, we used native mass spectrometry to analyze our purified RuvB. We are surprised to find that RuvB, which we used to assemble the complex, has endogenous ADP (Fig. S9D, also see below). Therefore, we believe that ADP molecules at protomers E and F were not replaced by ATP γ S while ADP molecules in protomer A were partially replaced by ATP γ S. We have revised our manuscript to clarify this point (see below).

“To rule out the ADP molecules in protomers E and F are endogenous or generated by ATP γ S hydrolysis, we did native mass spectrometry analysis of our purified RuvB and found that majority of our purified RuvB bind endogenous ADP molecules (Fig. S9D). Thus, the ADP molecules in protomers E and F were not exchanged into ATP γ S during our assembly of RuvB-DNA complex, suggesting that RuvBs at positions E and F prefer to bind ADP over ATP γ S.”

“The RuvB hexamer has approximate dimensions of 80 Å × 127 Å × 127 Å (Fig. 1B & 1C).” should refer to (Fig. 1B & 1D).

Thank you for pointing this out and we have corrected this point.

Figure 2E presents a reasonable mechanism of RuvB hexamer assembly. However, with little data to support the model (at least in the present communication), it should be presented as a “proposed” mechanism of hexamer assembly.

We have toned down our statement and switched to “a proposed mechanism of RuvB hexamer assembly” (see below).

Figure 2, panel D label has a typo. DNA-biding should be DNA-binding.

We thank the reviewer and have corrected this typo (see above).

1st paragraph of “DNA Binding by RuvB” section states “at the end of the HJ core-distal end.” This sentence should be edited since the complex is assembled without a HJ. Similarly, the last sentence of discussion could more accurately indicate that DNA (as opposed to HJ) is predicted to occur at 2 nts per step.

We have deleted the phrase of “at the end of the HJ core-distal end” and revised our last sentence of the discussion. Below are the revised statements.

“Specifically, three conserved arginine residues R297, R300, and R302 in the RuvB CTD form a motif known as an arginine finger to interact with the negatively charged DNA backbone.”

“Thus, the concerted events of nucleotide cycling, RuvB conformational oscillation, DNA pulling and rotating, as well as RuvA association and detachment ensure a precise DNA migration by a step size of 2 nucleotides.”

1st paragraph of “Nucleotide Cycling” has a typo at “the six promoters of RuvB hexamer”: promoters should be protomers. Same with the next sentence “Promoters B, C, and D bind ATP.....”

We thank the reviewer and have corrected these typos.

References:

- 1 Kouta Mayanagi and Yoshie Fujiwara and Tomoko Miyata and Kosuke, M. Electron microscopic single particle analysis of a tetrameric RuvA/RuvB/Holliday junction DNA complex. *Biochemical and biophysical research communications* **365**, 273-278, doi:<https://doi.org/10.1016/j.bbrc.2007.10.165> (2008).
- 2 Dennis, C., Fedorov, A., Käs, E., Salomé, L. & Grigoriev, M. RuvAB-directed branch migration of individual Holliday junctions is impeded by sequence heterology. *The EMBO Journal* **23**, 2413-2422, doi:10.1038/sj.emboj.7600249 (2004).

- 3 Ohnishi, T., Hishida, T., Harada, Y., Iwasaki, H. & Shinagawa, H. Structure-
Function Analysis of the Three Domains of RuvB DNA Motor Protein. *Journal of*
4 *Biological Chemistry* **280**, 30504-30510, doi:10.1074/jbc.m502400200 (2005).
- 5 Stephen, C. W. The RuvABC proteins and Holliday junction processing in
Escherichia coli. *Journal of Bacteriology* **178**, 1237-1241 (1996).
- 6 Mitchell, A. H. & West, S. C. Hexameric Rings of Escherichia coli RuvB Protein.:
Cooperative Assembly, Processivity and ATPase Activity. *Journal of molecular*
7 *biology* **243**, 208-215 (1994).
- 8 Adams, D. E. & West, S. C. Unwinding of Closed Circular DNA by the Escherichia
coli RuvA and RuvB Recombination/Repair Proteins. *Journal of molecular biology*
9 **247**, 404-417 (1995).
- 10 Yamada, K. *et al.* Crystal Structure of the RuvA-RuvB Complex. *Molecular Cell*
10, 671-681, doi:10.1016/s1097-2765(02)00641-x (2002).
- Ariyoshi, M., Nishino, T., Iwasaki, H., Shinagawa, H. & Morikawa, K. Crystal
structure of the Holliday junction DNA in complex with a single RuvA tetramer.
Proceedings of the National Academy of Sciences **97**, 8257-8262 (2000).
- Tsaneva, I. R., Müller, B. & West, S. C. RuvA and RuvB proteins of Escherichia
coli exhibit DNA helicase activity in vitro. *Proceedings of the National Academy of*
Sciences **90**, 1315-1319 (1993).
- George, H. *et al.* RuvAB-mediated branch migration does not involve extensive
DNA opening within the RuvB hexamer. *Curr Biol* **10**, 103-106,
doi:10.1016/s0960-9822(00)00296-7 (2000).

REVIEWERS' COMMENTS

Reviewer #1 (Remarks to the Author):

The authors have successfully addressed most of my concerns. I am satisfied with the revision. Two minor issues remain:

1. Page 4, the end of Introduction: "Together, we demonstrate that HJ migration includes a series of highly coordinated events, including nucleotide cycling, RuvB conformational changes that trigger DNA pulling and revolving, as well as the association and detachment of RuvA to RuvB." Though the authors captured different conformations of RuvB protomers in the structure, the stepwise mechanism demands more study. The structure is consistent or indicative of such highly coordinated events, but "demonstrate" is too strong a word here.

2. Page 8, towards the end of the page: "since ADP can be nicely fitted into the density with a sigma value of 1.8 while ATP can also be nicely placed into the density when we raised the sigma value to 2.5 (Fig. 5D)". While Fig. 5D legend states "ADP or ATP fitted to the cryo-EM density maps in protomer A at 2.5 sigma or 2.0 sigma, respectively." Either the text or the legend should be revised.

Reviewer #2 (Remarks to the Author):

I accept the response by the authors, and their arguments that the work by Rish et al complements the information about the structure of the RuvAB-HJ complex recently determined by other authors.

Reviewer #3 (Remarks to the Author):

The authors have added more experimental data and revised the text to address previous questions and concerns. There is just one lingering bit of confusion that should be addressed prior to publication:

At the top of page 9, the authors use mass spectrometry to identify ADP almost exclusively in the assembled RuvB complex that was analyzed structurally, even though ATPgS was added in vast excess to assemble the hexamer complex. The authors, thus, conclude that RuvB binds endogenous ADP and ATPgS is not exchanged. This finding begs the question of whether addition of ATPgS is needed for RuvB hexamer assembly. Particularly since endogenous ADP seems to be sufficient for doing so, it would seem that the RuvB hexamer should be isolated directly from the expression system with no assembly step needed. Since nucleotide identify is a key point of the mechanism presented in the study, the authors should comment (or test) whether additional nucleotide is needed (or not) for hexamer assembly.

Minor:

pg 9: "E115 was supposed to but did not form a salt-bridge with R205 in the AF dimer" should be reworded. E115 might be 'predicted' or 'hypothesized' to form such a salt bridge.

REVIEWERS' COMMENTS

Reviewer #1 (Remarks to the Author):

The authors have successfully addressed most of my concerns. I am satisfied with the revision. Two minor issues remain:

1. Page 4, the end of Introduction: "Together, we demonstrate that HJ migration includes a series of highly coordinated events, including nucleotide cycling, RuvB conformational changes that trigger DNA pulling and revolving, as well as the association and detachment of RuvA to RuvB." Though the authors captured different conformations of RuvB protomers in the structure, the stepwise mechanism demands more study. The structure is consistent or indicative of such highly coordinated events, but "demonstrate" is too strong a word here. We thank reviewer #1 for their positive comments and have made the following change to avoid making claims that are not fully supported by our results. "Together, we propose that HJ migration includes a series of highly coordinated events..."

2. Page 8, towards the end of the page: "since ADP can be nicely fitted into the density with a sigma value of 1.8 while ATP can also be nicely placed into the density when we raised the sigma value to 2.5 (Fig. 5D)". While Fig. 5D legend states "ADP or ATP fitted to the cryo-EM density maps in protomer A at 2.5 sigma or 2.0 sigma, respectively." Either the text or the legend should be revised.

We have revised the legend for Fig. 5D to align with the manuscript sigma threshold of 1.8. "(D) ADP or ATP fitted to the cryo-EM density maps in protomer A at 2.5 \$\sigma\$ or 1.8 \$\sigma\$, respectively."

Reviewer #2 (Remarks to the Author):

I accept the response by the authors, and their arguments that the work by Rish et al complements the information about the structure of the RuvAB-HJ complex recently determined by other authors.

We thank Review #2 for their comments and suggestions to improve the quality of our paper and we are pleased the reviewer accepts our responses.

Reviewer #3 (Remarks to the Author):

The authors have added more experimental data and revised the text to address previous questions and concerns. There is just one lingering bit of confusion that should be addressed prior to publication:

At the top of page 9, the authors use mass spectrometry to identify ADP almost exclusively in the assembled RuvB complex that was analyzed structurally, even though ATPgS was added in vast excess to assemble the hexamer complex. The authors, thus, conclude that RuvB binds

endogenous ADP and ATPgS is not exchanged. This finding begs the question of whether addition of ATPgS is needed for RuvB hexamer assembly. Particularly since endogenous ADP seems to be sufficient for doing so, it would seem that the RuvB hexamer should be isolated directly from the expression system with no assembly step needed. Since nucleotide identity is a key point of the mechanism presented in the study, the authors should comment (or test) whether additional nucleotide is needed (or not) for hexamer assembly.

We thank Reviewer #3 for their meticulous attention to detail to address whether ATPgS is needed for RuvB hexamer assembly. To address this concern, we assembled RuvB with DNA substrate (as described in the manuscript) in the absence of ATPgS and demonstrated its ability to oligomerize via gel filtration. As seen in the figure below, RuvB hexamer did not assemble in the absence of additional nucleotides, necessitating an assembly step with additional nucleotides. Panel (a) shows an S200 10/300 increase column gel filtration curve of purified RuvB that was prepared with HJ DNA substrate, in the absence of ATPgS. The RuvB monomer was verified by SDS-Page in panel (b) with “SL” being the sample that was loaded onto the column as a control. Additionally, HJ DNA substrate was not detected in the RuvB monomer peak as seen in the agarose gel stained with ethidium bromide, in panel (c).

Minor:

pg 9: “E115 was supposed to but did not form a salt-bridge with R205 in the AF dimer” should be reworded. E115 might be ‘predicted’ or ‘hypothesized’ to form such a salt bridge. Since the E115 & R205 salt bridge interaction was computationally predicted we have amended the above statement to avoid misleading readers. “E115 was predicted to but did not form a salt-bridge with R205 in the AF dimer;”.